

**Long term variations of actual evapotranspiration over the Tibetan**
**Plateau**
Cunbo Han [1,2], Yaoming Ma[1,3,4,5], Binbin Wang[1], Lei Zhong[6], Weiqiang Ma[1,3],
Xuelong Chen[1,3], Zhongbo Su[7]
1.  Key Laboratory of Tibetan Environment Changes and Land Surface
Processes, Institute of Tibetan Plateau Research, Chinese Academy of
Sciences, Chinese Academy of Sciences, Beijing, China
2.  Institute for Meteorology and Climate Research, Karlsruhe Institute of
Technology, Karlsruhe, Germany
3.  CAS Center for Excellence in Tibetan Plateau Earth Sciences, Chinese
Academy of Sciences, Beijing, China
4.  University of Chinese Academy of Sciences, Beijing, China
5.  Lanzhou University, Lanzhou, China
6.  Laboratory for Atmospheric Observation and Climate Environment
Research, School of Earth and Space Sciences, University of Science
and Technology of China, Hefei, China
7.  Faculty of Geo-Information Science and Earth Observation, University of
Twente, Enschede, The Netherlands
**Correspondence to:**
Prof. Dr. Yaoming Ma
Institute of Tibetan Plateau Research, Chinese Academy of Sciences
(ITPCAS)
16-3 Lincui Road, Chaoyang District, Beijing
100101, China
Tel: +86 010 84097079
Email: ymma@itpcas.ac.cn



**Abstract**
Terrestrial actual evapotranspiration ($ET_a$) is a key parameter controlling the
land-atmosphere interaction processes and the water cycle. However, the
spatial distribution and temporal changes of $ET_a$ over the Tibetan Plateau (TP)
remain very uncertain. Here we estimate the multiyear (2001-2018) monthly
$ET_a$ and its spatial distribution on the TP by a combination of meteorological
data and satellite products. Validation against data from six eddy-covariance
monitoring sites yielded a root mean square errors ranging from 9.3 to 14.5
mm mo$^{-1}$, and correlation coefficients exceeding 0.9. The domain mean of
annual $ET_a$ on the TP decreased slightly (-1.45 mm yr$^{-1}$, $p < 0.05$) from 2001
to 2018. The annual $ET_a$ increased significantly at a rate of 2.62 mm yr$^{-1}$ ($p <$
0.05) in the eastern sector of the TP (lon > 90° E), but decreased significantly
at a rate of -5.52 mm yr$^{-1}$ ($p < 0.05$) in the western sector of the TP (lon < 90°
E). In addition, the decreases in annual $ET_a$ were pronounced in spring and
summer seasons, while almost no trends were detected in the autumn and
winter seasons. The mean annual $ET_a$ during 2001-2018 and over the whole
TP was 496 ± 23 mm. Thus, the total evapotranspiration from the terrestrial
surface of the TP was 1238.3 ± 57.6 km$^3$ yr$^{-1}$. The estimated $ET_a$ product
presented in this study is useful for an improved understanding of changes in
energy and water cycle on the TP. The dataset is freely available at the
Science Data Bank (http://www.dx.doi.org/10.11922/sciencedb.t00000.00010,
(Han et al. 2020)) and at the National Tibetan Plateau Data Center
(https://data.tpdc.ac.cn/en/data/5a0d2e28-ebc6-4ea4-8ce4-a7f2897c8ee6/).

**Key words**: Actual evapotranspiration; SEBS; Tibetan Plateau; Trend.




**Key points:**
- The SEBS-estimated monthly $ET_a$ during 2001-2018 shows acceptable
accuracy validated against 6 flux towers.
- Annual $ET_a$ over the entire TP and in the western TP decrease
significantly, while it increases in the east TP.
- Decrease of annual $ET_a$ is pronounced in spring and summer, while
almost no trends are detected in autumn and winter.




## 1    Introduction


As the birthplace of Asia's major rivers, the Tibetan Plateau (TP), famous as
the "Water Tower of Asia", is essential to the Asian energy and water cycles
(Immerzeel et al. 2010, Yao et al. 2012). Along with increasing air
temperature, evidence from the changes of precipitation, runoff, and soil
moisture indicates that the hydrological cycle of the TP has been intensified
during the past century (Yang et al. 2014). Contributing around two-thirds of
global terrestrial precipitation, evapotranspiration ($ET$) is a crucial component
that affects the exchange of water and energy between the land surface and
the atmosphere (Oki and Kanae 2006, Fisher et al. 2017). $ET$ is also an
essential factor modulating regional and global weather and climate. As the
only connecting component between the energy budget and the water cycle in
the terrestrial ecosystems (Xu and Singh 2005), $ET$ and variations of $ET$ over
the TP have received increasing attention worldwide (Xu and Singh 2005, Li
et al. 2014, Zhang et al. 2018, Yao et al. 2019, Wang et al. 2020). Total
evaporation from large lakes of the TP has been quantitatively estimated
recently (Wang et al. 2020), however, the terrestrial $ET$ on the TP and its
spatial and temporal changes remain very uncertain.

Many studies have tried to evaluate $ET$'s temporal and spatial variability
across the TP using various methods. The pan evaporation ($E_{pan}$), that
represents the amount of water evaporated from an open circular pan, is the
most popular observational data source of $ET$. Long time series of $E_{pan}$ are
often available with good comparability among various regional
measurements. Thus, it has been widely used in various disciplines, e.g.,
meteorology, hydrology, and ecology. Several studies have revealed the trend
of $E_{pan}$ on the TP (Zhang et al. 2007, Liu et al. 2011, Shi et al. 2017, Zhang et
al. 2018, Yao et al. 2019). Although $E_{pan}$ and potential $ET$ suggest the long-



term variability of $ET$ according to contrasting trends between $E_{pan}$ and actual
$ET$ ($ET_a$) (Zhang et al. 2007), these measures cannot precisely depict the
spatial pattern of trends in $ET_a$. Recently, several studies applied revised
models, which are based on the complementary relationship (CR) of $ET$, to
estimate $ET_a$ on the TP (Zhang et al. 2018, Ma et al. 2019, Wang et al. 2020).
Employing only routine meteorological observations without requiring any
vegetation and soil information is the most significant advantage of CR
models (Szilagyi et al. 2017). However, numerous assumptions and
requirements of validations of key parameters limit the application and
performance of CR models over different climate conditions. The application
of eddy-covariance (EC) technologies in the past decade has dramatically
advanced our understanding of the terrestrial energy balance and $ET_a$ over
various ecosystems across the TP. However, the fetch of the EC observation
is on the order of hundreds of meters, thus impeding the ability to capture the
plateau-scale variations of $ET_a$. Therefore, finding an effective way to advance
the estimation of $ET_a$ on the TP is of great importance.

Satellite remote sensing (RS) provides temporally frequent and spatially
contiguous measurements of land surface characteristics that affect $ET$, for
example, land surface temperature, albedo, vegetation index. Satellite RS
also offers the opportunity to retrieve $ET$ over a heterogeneous surface
(Zhang et al. 2010). Multiple RS-based algorithms have been proposed.
Among these algorithms, the surface energy balance system (SEBS)
proposed by Su (2002) has been widely applied to retrieve land surface
turbulent fluxes on the TP (Chen et al. 2013, Ma et al. 2014, Han et al. 2016,
Han et al. 2017, Zou et al. 2018, Zhong et al. 2019). Chen et al. (2013)
improved the roughness length parameterization scheme for heat transfer in
SEBS to expand its modeling applicability over bare ground, sparse canopy,



dense canopy, and snow surfaces in the TP. An algorithm for effective
aerodynamic roughness length had been introduced into the SEBS model to
parameterize subgrid-scale topographical form drag (Han et al. 2015, Han et
al. 2017). This scheme improved the skill of the SEBS model in estimating the
surface energy budget over mountainous regions of the TP. A recent advance
by Chen et al. (2019) optimized five critical parameters in SEBS using
observations collected from 27 sites globally, including 6 sites on the TP, and
suggested that the overestimation of the global $ET$ was substantially improved
with the use of optimal parameters.

While the spatial and temporal pattern of the $ET_a$ in the TP had been
investigated in many studies (Zhang et al. 2007, Zhang et al. 2018, Wang et
al. 2020), considerable inconsistencies for both trends and magnitudes of $ET_a$
exist due to uncertainties in forcing and parameters used by various models.
Thus, in this study, with full consideration of the recent developments in the
SEBS model over the TP, we aim to (1) develop an 18-year (2001-2018) $ET_a$
product of the TP, along with independent validations against EC
observations; (2) quantify the spatiotemporal variability of the $ET_a$ in the TP,
and (3) uncover the main factors dominating the changes in $ET_a$, using the
estimated product.

**2  Methodology and data**
**2.1  Model description**
The SEBS model (Su 2002) was used to derive land surface energy flux
components in the present study. The remote-sensed land surface energy
balance equation is given by



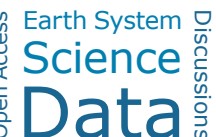

$$R_n = H + LE + G_0. \qquad (1)$$

$R_n$ is the net radiation flux (W m⁻²), $H$ is the sensible heat flux (W m⁻²), $LE$ is
the latent heat flux (W m⁻²), and $G_0$ is the ground heat flux (W m⁻²).

The land surface net radiation flux was computed as
$$R_n = (1 - \alpha) \times SWD + LWD - \varepsilon \times \sigma \times T_s^4 \qquad (2)$$

where $\alpha$ is the land surface albedo derived from the Moderate Resolution
Imaging Spectroradiometer (MODIS) products. Downward shortwave (*SWD*)
and longwave (*LWD*) radiation were obtained from the China Meteorological
Forcing Dataset (CMFD). Land surface temperature ($T_s$) and emissivity ($\varepsilon$)
values were also obtained from MODIS products.

In vegetated areas the soil heat flux, $G_0$, was calculated from the net radiation
flux and vegetation cover
$$G_0 = R_n \times (r_c \times f_c + r_s \times (1 - f_c)). \qquad (3)$$

$r_s$ and $r_c$ are ratios of ground heat flux and net radiation for surfaces with bare
soil and full vegetation, respectively. Fractional vegetation cover ($f_c$) was
derived from the normalized difference vegetation index (NDVI). Over water
surfaces (NDVI < 0 and $\alpha$ < 0.47), $G_0 = 0.5R_n$ was used (Gao et al. 2011,
Chen et al. 2013). On glaciers, $G_0$ is negligible (Yang et al. 2011) and $G_0$ =
0.05$R_n$.

In the atmospheric surface layer, sensible heat flux and friction velocity were
calculated based on the Monin-Obukhov similarity (Stull 1988),
$$U = \frac{u_*}{\kappa} \left[ ln\left(\frac{z - d_0}{z_{0m}^{eff}}\right) - \psi_m\left(\frac{z - d_0}{L}\right) + \psi_m(\frac{z_{0m}^{eff}}{L}) \right] \qquad (4)$$

$$\theta_0 - \theta_a = \frac{H}{\kappa u_* \rho C_p} \left[ ln\left(\frac{z - d_0}{z_{0h}^{eff}}\right) - \psi_h\left(\frac{z - d_0}{L}\right) + \psi_h(\frac{z_{0h}^{eff}}{L}) \right] \qquad (5)$$

$$L = \frac{\rho C_p u_*^3 \theta_v}{\kappa g H}. \qquad (6)$$



$U$ is the horizontal wind velocity at a reference height $z$ (m) above the ground
surface, $\theta_0$ is the potential temperature at the land surface (K), $\theta_a$ is the
potential temperature (K) at the reference height $z$, $d_0$ is the zero-plane
displacement height (m), $\rho$ is the air density (kg m$^{-3}$), $C_p$ is the specific heat for
moist air (J kg$^{-1}$ °C$^{-1}$), $\kappa = 0.4$ is the von Kármán's constant, $u_*$ is the friction
velocity, $L$ is the Monin-Obukhov length (m), $\theta_v$ is the potential virtual
temperature (K) at the reference height $z$, $\psi_m$ and $\psi_h$ are the stability
correction functions for momentum and sensible heat transfer respectively,
and $g$ is the gravity acceleration (m s$^{-2}$). To account for the form drag caused
by subgrid-scale topographical obstacles, effective roughness lengths for
momentum ($z_{0m}^{eff}$, m) and sensible heat ($z_{0h}^{eff}$, m) transfer were introduced into
the SEBS model by Han et al. (2017). These modifications are parameterized
as follows (Grant and Mason 1990, Han et al. 2015),
$$ln^2\left(h/2z_{0m}^{eff}\right) = \frac{\kappa^2}{0.5D\lambda + \kappa^2/ln^2(h/2z_{0m})} \tag{7}$$

$$ln\left(h/2z_{0h}^{eff} + 1\right) = ln\left(h/2z_{0h} + 1\right)\frac{ln\,(h/2z_{0m}+1)}{ln\,(h/z_{0m}^{eff}+1)} \tag{8}$$

where $\lambda$ is the average density of the subgrid-scale roughness elements
calculated from digital elevation models, $D$ is the form drag coefficient and
$D=0.4$ is used for the mountainous areas of the TP as suggested by Han et al.
(2015), $z_{0m}$ and $z_{0h}$ are the local-scale roughness lengths for momentum (m)
and heat transfer (m), respectively. Detailed calculations can be found in Su
(2002). A revised algorithm for $z_{0h}$ developed by Chen et al. (2013) was
applied as this algorithm outperforms the original scheme of the SEBS model
on the TP.

To constrain the actual evapotranspiration, an evaporative fraction was
applied in the SEBS model. Under the dry-limit condition, the evaporation
becomes zero due to the limited supply of available soil moisture, while water
vapor evaporates at the potential rate under the wet-limit condition (Su 2002).



Finally, daily $ET_a$ was calculated using the evaporative fraction as a residual of
the surface energy budget equation while accounting for dry and wet limits.
Details are available in Su (2002).
**2.2 Data**
In-situ observations, satellite-based products, and meteorological forcing data
were used in this study to estimate monthly $ET_a$ over the TP area. The CMFD,
that was developed based on the released China Meteorological
Administration (CMA) data (He et al. 2020), was used as model input. The
CMFD covers the whole landmass of China at a spatial resolution of 0.1° and
a temporal resolution of three hours. The dataset was established through the
fusion of in-situ observations, remote sensing products, and reanalysis
datasets. In particular, the dataset benefits from the merging of the
observations at about 700 CMA's weather stations, and by using the Global
Energy and Water Cycle Experiment – Surface Radiation Budget (GEWEX-
SRB) shortwave radiation dataset (Pinker and Laszlo 1992). The GEWEX-
SRB data has not been used in any other reanalysis dataset. In addition,
independent datasets observed in western China where weather stations are
scarce were used to evaluate the CMFD. This includes data collected through
the Heihe Watershed Allied Telemetry Experimental Research (HiWATER) (Li
et al. 2013) and the Coordinated Enhanced Observing Period (CEOP) Asia-
Australia Monsoon Project (CAMP) (Ma et al. 2003). CMFD dataset is suitable
for our study due to its continuous-time coverage and consistent quality.
Detailed information for the CMFD dataset is listed in Table 1.

In-situ EC data observed at six flux stations on the TP were used to validate
model results. Locations of the six observation sites are illustrated in Figure 1
and detailed descriptions for these six sites are shown in Table 2. The



instrumental setup at each site consists of: an EC system comprising a sonic
anemometer (CSAT3, Campbell Scientific Inc) and an open-path gas analyzer
(LI-7500, Li-COR); a four-component radiation flux system (CNR-1, Kipp &
Zonen), installed at a height of 1.5 m; a soil heat flux plate (Hukseflux,
HFP01), buried in the soil to a depth of 0.1 m; soil moisture and temperature
probes, buried at a depth of 0.05, 0.10, and 0.15 m, respectively (Han et al.
2017). The EC data were processed with the EC software package TK3
(Mauder and Foken 2015). The main post-processing procedures were as
follows: spike detection, coordinate rotation, spectral loss correction,
frequency response corrections (Moore 1986), and corrections for density
fluctuations (Webb et al. 1980). The ground heat flux was obtained by
summing the flux value observed by the heat flux plate and the energy
storage in the layer above the heat flux plate (Han et al. 2016). Monthly EC
data, which are used for validation, were generated from half-hourly variables.
A more comprehensive dataset including the EC data used in this work has
been published and is freely available (Ma et al. 2020).
**2.3   Model evaluation metrics and data analysis methods**
The model performance was assessed using the Pearson correlation
coefficient (R), the root mean square error (RMSE), and the mean bias (MB)
between the estimated and observed monthly $ET_a$ at the six stations on the
TP.

The least-square regression technique was used to detect the long-term linear
annual trends in $ET_a$ values. The linear model to simulate $ET_a$ values ($Y_t$)
against time ($t$) is
$$Y_t = Y_0 + bt + \varepsilon_t \tag{9}$$



The Student's *t*-test, having an *n*-2 degree of freedom (*n* is the number of
samples), was used to evaluated the statistical significance of the linear
trends, and only tests with a *p*-value less than 0.05 were selected as having
passed the significance test.
**3  Results and discussion**
**3.1  Validation against flux tower observations**
The SEBS-estimated $ET_a$ was validated against EC observations at the six
flux stations on the TP at a monthly scale (Figure 2). The SEBS model is
capable of capturing both the magnitude and phase of the monthly $ET_a$ signal
at all the six stations. The correlation coefficients are all larger than 0.9 and
have passed the significance test at the $p$ = 0.01 level. RMSE values range
from 9.3 to 14.5 mm mo$^{-1}$ with the minimum at the BJ station and the
maximum at the SETORS station. The MB values are all negative except at
the NADORS station, which means the SEBS model slightly underestimated
$ET_a$ values on the TP.

Specifically, the SEBS model performed particularly well at the spare grass
stations (NADORS and MAWORS) and at the short grass sites (BJ and
NAMORS). At the high grass site (SETORS) and the gravel site (QOMS), the
SEBS model is capable of reproducing the EC-observed monthly $ET_a$ with
RMSE values of 14.5 and 13.2 mm mo$^{-1}$, respectively. In addition, the
underestimates of $ET_a$ by SEBS are mostly in the dry season, when the
canopy is withered. The validation at the site-scale indicates that the SEBS
model used in this work can be applied to a wide range of ecosystems over
the TP.



## 3.2 Spatial distribution

There was a clear spatial pattern to the multiyear (2001-2018) mean annual $ET_a$ (Figure 3). In general, the SEBS-estimated $ET_a$ decreases from the southeast to the northwest of the TP, with the maximum value above 1200 mm in the southeastern Tibet and Hengduan Mountains and the minimum value less than 100 mm in the northwestern edge of the TP. In the central TP, where there are several lakes, $ET_a$ was typically from 500 to 1000 mm. $ET_a$ was lower than 200 mm over the high, snow- and ice-bound, mountainous areas. For example, over the northern slopes of the Himalaya, Nyenchen Tanglha Mountains, and the eastern section of the Tanggula Mountains. The reason is that these snow- and ice-bound mountainous areas have a higher ability to reflect downward shortwave radiation and hence have less available energy to evaporate. On the whole, the domain averaged multiyear mean annual $ET_a$ over the TP is 496±23 mm. The total amount of water evapotranspirated from the terrestrial surface of the TP are around 1238.3±57.6 $km^3$ $yr^{-1}$, considering the area of the TP to be $2.5×10^6$ $km^2$.

Figure 4 shows the multi-year average spring (Marth, April, and May), summer (June, July, and August), autumn (September, October, and November), and winter (December, January, and February) $ET_a$ on the TP. Generally, the distribution pattern of seasonal $ET_a$ was comparable with that of the annual $ET_a$. Both seasonal and annual $ET_a$ show a decreasing trend from the southeastern TP to the northwestern TP. Note that the distribution pattern almost faded out in winter season owing to a minimum in available energy and precipitation (Figure 4d). The $ET_a$ in spring is higher than that in autumn, except for some high mountainous areas (e.g.: mountain ranges of Himalaya and Hengduan mountains). The spring $ET_a$ ranges from 50 mm to 450 mm, while autumn $ET_a$ ranges from 50 mm to 250 mm. In summer, the $ET_a$ is





larger than 250 mm in most of the TP, while the $ET_a$ is still below 100 mm in
large areas of the northwestern TP. The multiyear seasonal $ET_a$ averaged
over the whole TP is 140±10 mm, 256±12 mm, 84±5 mm, and 34±4 mm, for
spring, summer, autumn, and winter, respectively.
**3.3  Trend analysis**
The trend of annual $ET_a$ during 2001-2018 is shown in Figure 5. Overall, an
increasing trend of SEBS-simulated $ET_a$ is dominant in the eastern TP (lon >
90° E) while a decreasing trend is dominant in the western TP (lon < 90° E).
The trends pass the $t$-test ($p < 0.05$) in most part of the areas. The decreasing
trend in the western TP is pronounced and passes the $t$-test ($p < 0.05$). This
trend is larger than -7.5 mm yr$^{-1}$ in most parts of the area and even larger than
-10 mm yr$^{-1}$ in a few parts. In the eastern TP, the increasing trend is mostly
between 5 and 10 mm yr$^{-1}$ and passes the $t$-test ($p < 0.05$). The $ET_a$ trend
tends to be greater along the marginal region of the northern, eastern, and
southeastern TP. Along the marginal region of the southwestern TP and in the
western section of Himalaya Mountains this trend weakens.

The trends of seasonal $ET_a$ between 2001 and 2018 are spatially
heterogeneous over the TP (Figure 6). Decreasing trends in spring and
summer are generally at a rate between -2.5 and -7.5 mm yr$^{-1}$, and increasing
trends are generally at a rate below 5.0 mm yr$^{-1}$ and 7.5 mm yr$^{-1}$ in spring and
summer, respectively. Areas showing decreasing $ET_a$ tend to become larger in
autumn and winter seasons. Both the decreasing and increasing trends are
subdued in autumn and winter compared with that in spring and summer
seasons. Decreasing rates of $ET_a$ in autumn and winter are generally below -
2.5 mm yr$^{-1}$, and only a few areas have a rate larger than -2.5 mm yr$^{-1}$.

Due to the contrast in the trends in the eastern and western halves of the TP,
we divided the TP into two regions: the eastern TP (lon > 90° E) and the
western TP (lon < 90° E). Trends of the $ET_a$ anomaly averaged over the entire
TP, the western TP, and the eastern TP are shown in Figure 7a. The domain
means of $ET_a$ on the TP as a whole, and in the western TP decreased at rates
of -1.45 mm yr$^{-1}$ and -5.52 mm yr$^{-1}$, respectively. However, the $ET_a$ in the
eastern TP increased at a rate of 2.62 mm yr$^{-1}$. The decreasing rate of $ET_a$ in
the entire TP is influenced mainly by the significant decrease of $ET_a$ in the
western TP. Seasonally, the rates of change of $ET_a$ over the whole TP are -
0.82 mm yr$^{-1}$ ($p < 0.05$) and -0.79 mm yr$^{-1}$ ($p < 0.05$) in spring and summer,
respectively (Figure 7b). However, in autumn and winter the $ET_a$ changes at a
rate of 0.10 mm yr$^{-1}$ and 0.06 mm yr$^{-1}$, respectively, and do not pass the $t$-test
($p < 0.05$). $ET_a$ in spring and summer seasons account for 75.7% of the
annual $ET_a$. The variation in amplitude and changing rates in these two
seasons are much larger than in the other seasons. Moreover, spatial
distributions of spring and summer $ET_a$ trends are close to that of the annual
$ET_a$ trend (Figure 6). Thus, changes of $ET_a$ in the spring and summer
dominate the variations of $ET_a$ in the whole year.

The decrease of $ET_a$ over the whole TP and in the western TP during 2001-
2018 can be explained by the decrease of $R_n$ in the same time period (Figure
8a). From 2001 to 2012, $ET_a$ averaged over the entire TP increased slightly
and then decreased dramatically from 2012, reaching a minimum in 2014.
The significant decrease in $ET_a$ between 2012 and 2014 was due to the rapid
decline of the $R_n$ (Figure 8a). In the eastern TP, $ET_a$ increased during 2001-
2018, while $R_n$ decreased in the same period. Thus, $R_n$ was not the dominant
factor controlling the annual variations of $ET_a$. However, the increasing trends
of both precipitation and air temperature can explain the increase of $ET_a$ in the
eastern TP during the period 2001-2018 (Figure 8b and Figure 8c). The
increasing precipitation increased the water resource available for $ET_a$.
Moreover, the increasing air temperature accelerated the melting of
permafrost and glaciers on the TP. Hence, the melting water replenished the
ecosystem and increased the $ET_a$ of the eastern TP.

Although the domain-averaged trend in $ET_a$ has been decreasing across the
entire TP from 2001 to 2018, $ET_a$ values in some areas have increased.
Moreover, the changing rates also depend on the time series of $ET_a$. For
example, the $ET_a$ increased slightly from 2001 to 2012, while decreased from
2001 to 2018. This demonstrates the necessity to utilize high-spatial
resolution datasets and long time series to investigate the trends in $ET_a$ over
the TP.
**4   Summary and conclusions**
The SEBS-estimated $ET_a$ is at a resolution of around 10 km, while the
footprint of EC observed $ET_a$ values ranges from a few dozen meters to a few
hundreds of meters. SEBS-estimated $ET_a$ compares very well with
observations at the six flux towers, showing low RMSE and MB values. These
estimates were able to capture annual and seasonal variations in $ET_a$, despite
these two datasets being mismatched in their spatial representation.

Heterogeneous land surface characteristics and nonlinear changes in
atmospheric conditions resulted in heterogeneities in spatial distributions of
$ET_a$ and changes in $ET_a$. The SEBS-estimated multiyear (2001-2018) mean
annual $ET_a$ on the TP was 515±22 mm, resulting in approximately
1287.5±55.0 km$^3$ yr$^{-1}$ of total water evapotranspiration from the terrestrial
surface. Annual $ET_a$ generally decreased from the southeast to the northwest





of the TP. The maximum was over 1200 mm, in the southeastern Tibet and
Hengduan Mountains, while the minimum was less than 100 mm in the
northwest marginal area of the TP. Moreover, $ET_a$ was typically lower than 200
mm over snow- and ice-bound mountainous areas, as there was limited
available energy to evaporate the water.

Averaged over the entire TP, annual $ET_a$ increased slightly from 2001 to 2012,
but decreased significantly after 2012 and reached a minimum in 2014.
Generally, there was a slight decreasing trend in the domain mean annual $ET_a$
on the TP at the rate of -1.45 mm $yr^{-1}$ ($p < 0.05$) from 2001 to 2018. However,
trends of annual $ET_a$ were opposite in the western and eastern TP. The
annual $ET_a$ decreased significantly in the western TP at a rate of -5.52 mm $yr^{-1}$
$^{1}$ ($p < 0.05$) from 2001 to 2018, while annual $ET_a$ in the eastern TP increased
at a rate of 2.62 mm $yr^{-1}$ ($p < 0.05$) in the same period.

The spatial distributions of seasonal $ET_a$ trends were also noticeably
heterogeneous during 2001-2018. The spatial patterns of rate of change of
$ET_a$ in spring and summer were similar to the annual changes in $ET_a$. Finally,
$ET_a$ decreased as well in the spring and summer season but at slower rates
compared with the annual $ET_a$, however, only very weak trends were found in
the autumn and winter seasons.

**5   Data availability**
The dataset presented and analyzed in this article has been released and is
available for free download from the Science Data Bank
(http://www.dx.doi.org/10.11922/sciencedb.t00000.00010, (Han et al. 2020))
and from the National Tibetan Plateau Data Center



(https://data.tpdc.ac.cn/en/data/5a0d2e28-ebc6-4ea4-8ce4-a7f2897c8ee6/).
The dataset is published under the Creative Commons Attribution 4.0
International (CC BY 4.0) license.

**Acknowledgments**
This study was funded by the Second Tibetan Plateau Scientific Expedition
and Research (STEP) program (grant no. 2019QZKK0103), the Strategic
Priority Research Program of Chinese Academy of Sciences (XDA20060101),
the National Natural Science Foundation of China (91837208, 41705005, and
41830650). The CMFD data were obtained from the National Tibetan Plateau
Data Center (https://data.tpdc.ac.cn/en/data/8028b944-daaa-4511-8769-
965612652c49/). MODIS data were obtained from the NASA Land Processes
Distributed Active Archive Center (https://lpdaac.usgs.gov/). Global 1 km
forest canopy height data were obtained from the Oak Ridge National
Laboratory Distributed Active Archive Center for Biogeochemical Dynamics
(https://daac.ornl.gov/cgi-bin/dsviewer.pl?ds_id=1271). The authors would like
to thank all colleagues working at the observational stations on the TP for their
maintenance of the instruments.

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





**List of tables**





Table 1: Input datasets used in this study.

| Variables | Data source | Availability | Temporal resolution | Spatial resolution |
|-----------|-------------|--------------|---------------------|--------------------|
| Downward Shortwave | CMFD | 1979 – 2018 | 3 hours | 0.1° |
| Downward longwave | CMFD | 1979 – 2018 | 3 hours | 0.1° |
| Air temperature | CMFD | 1979 – 2018 | 3 hours | 0.1° |
| Specific humidity | CMFD | 1979 – 2018 | 3 hours | 0.1° |
| Wind velocity | CMFD | 1979 – 2018 | 3 hours | 0.1° |
| Land surface temperature | MOD11C3 | 2001 – now | Monthly | 0.05° |
| Land surface emissivity | MOD11C3 | 2001 – now | Monthly | 0.05° |
| Height of canopy | GLAS & SPOT | 2000 - now | Monthly | 0.01° |
| Albedo | MOD09CMG | 2001 - now | Daily | 0.05° |
| *NDVI* | MOD13C2 | 2001 - now | Monthly | 0.05° |
| DEM | ASTER GDEM | - | - | 30 m |








Table 2: Station information.

| Station | Location | Elevation (m) | Land cover |
|---------|----------|---------------|------------|
| QOMS | 28.21°N, 86.56°E | 4276 | Gravel |
| NAMORS | 30.46°N, 90.59°E | 4730 | Grassy marshland |
| SETORS | 29.77°N, 94.73°E | 3326 | Grass land |
| NADORS | 33.39°N, 79.70°E | 4264 | Sparse grass-Gobi |
| MAWORS | 38.41°N, 75.05°E | 3668 | Sparse grass-Gobi |
| BJ | 31.37°N, 91.90°E | 4509 | Sparseness meadow |




**List of figures**


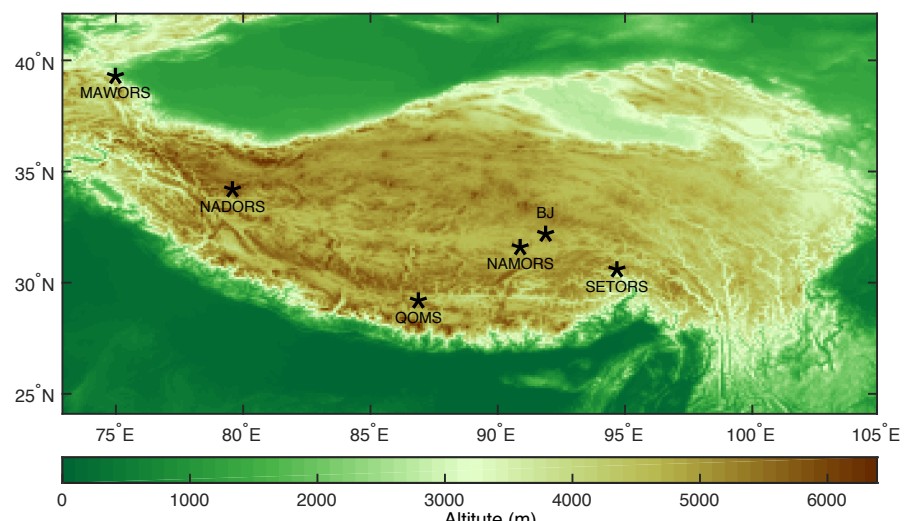


Figure 1: Locations of the six flux tower sites (marked by pentagrams) on the

TP. The legend of the color map is elevation above mean sea level in meters.





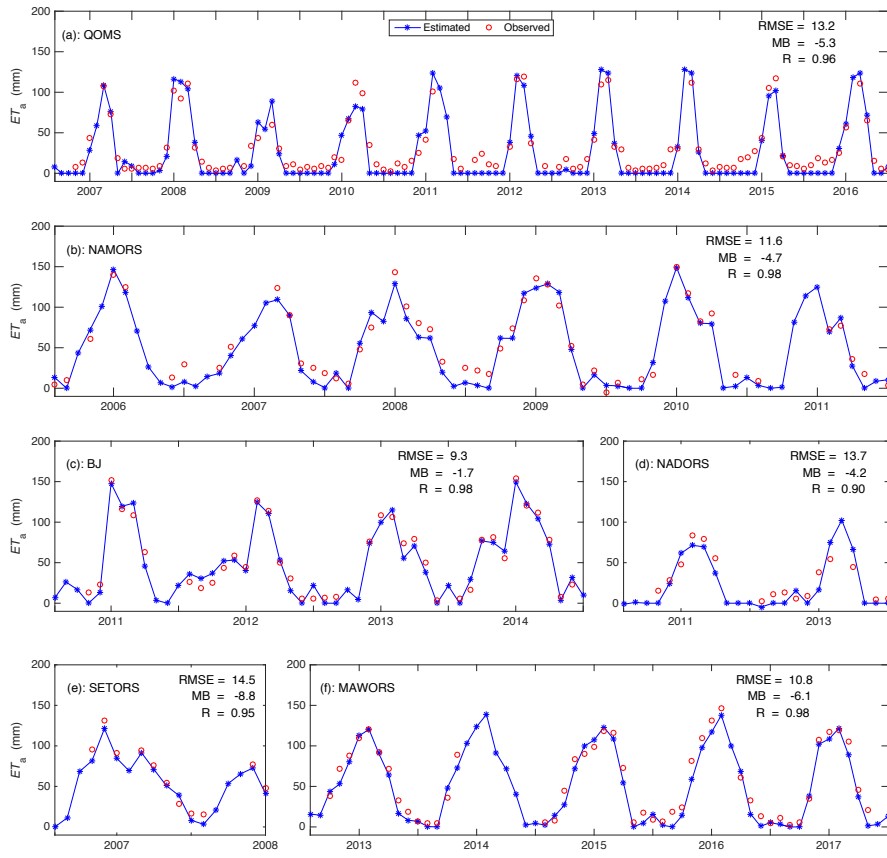


Figure 2: SEBS-estimated and EC-observed monthly $ET_a$ at the six stations
(a-f) on the TP in years when the latter observations were available. RMSE is
the root-mean-square error, MB is the mean bias, and R is the correlation
coefficient.






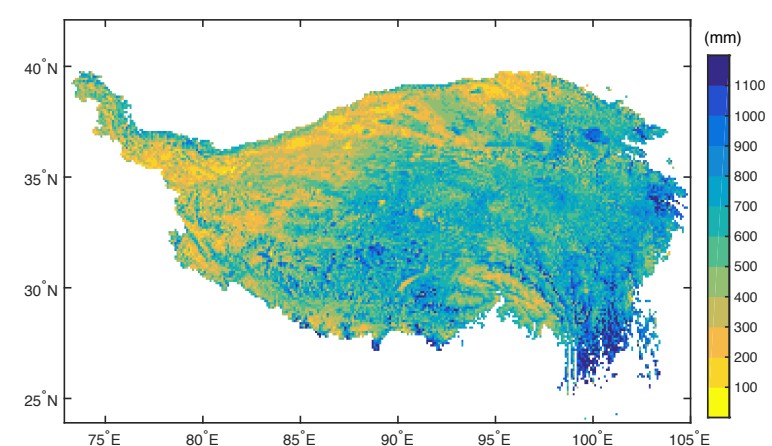


Figure 3: Spatial distribution of the SEBS-estimated multiyear (2001-2018)
average annual $ET_a$.


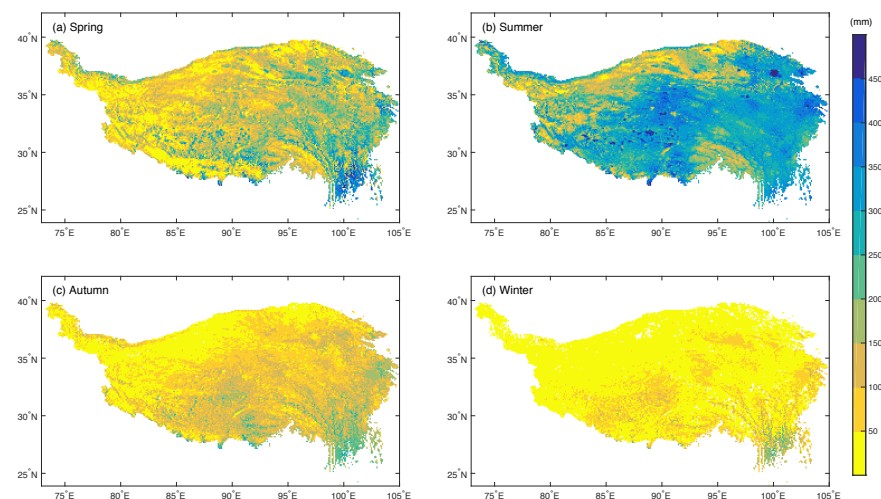


Figure 4: Spatial distributions of the SEBS-estimated multiyear (2001-2018)
average seasonal $ET_a$ (mm/season) values over the TP. (a) spring, (b)
summer, (c) autumn, (d) winter.




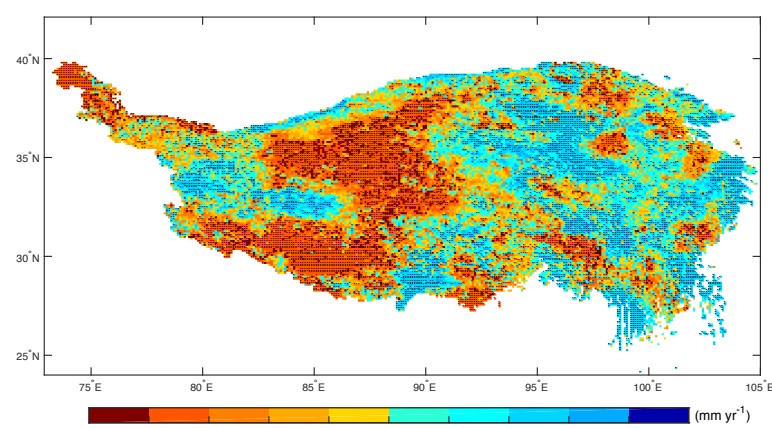


Figure 5: Spatial distribution of annual $ET_a$ linear trend on the TP from 2001 to
2018. The stippling indicates the trends that pass the t-test ($p < 0.05$).




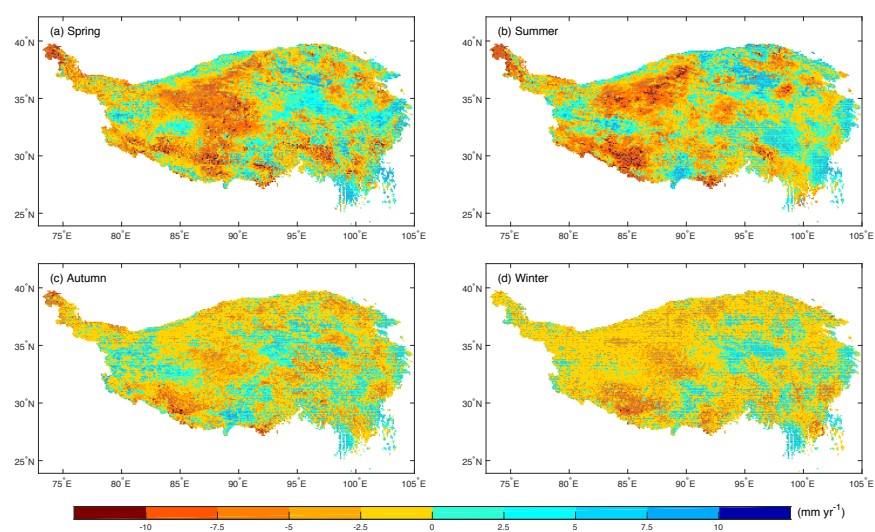


Figure 6: Spatial distributions of seasonal $ET_a$ linear trends on the TP from
2001 to 2018: (a) annual, (b) spring, (c) summer, (d) autumn, (e) winter. The
stippling indicates the trends that pass the $t$-test ($p < 0.05$).


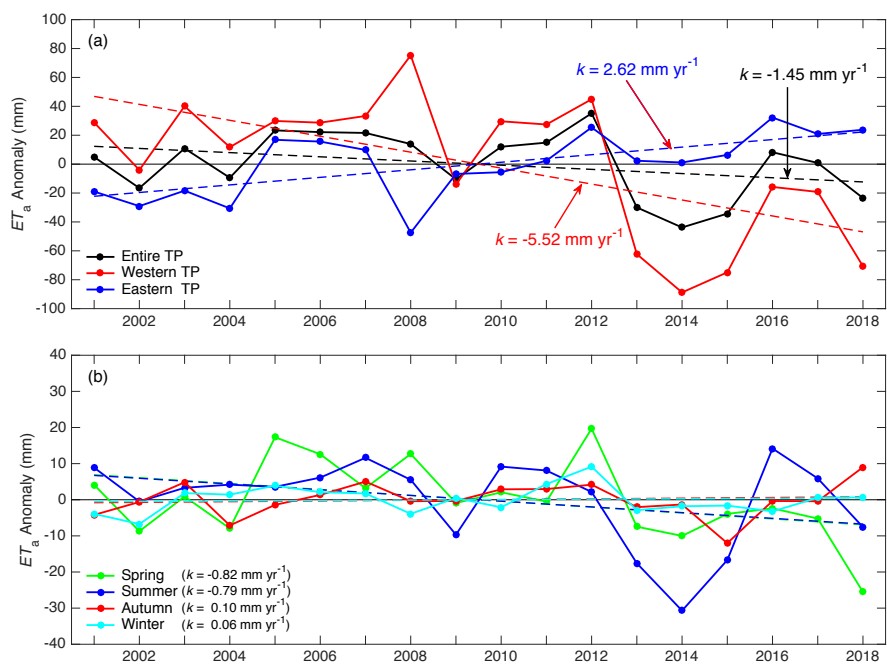


Figure 7: Anomalies of the domain-averaged annual $ET_a$ of the entire TP, the
western TP (lon < 90° E), and the eastern TP (lon > 90° E), respectively (a).
Domain-averaged seasonal $ET_a$ anomalies over the entire TP (b). The dashed
straight lines indicate linear trends during 2001-2018, and $k$ is the slope of the
straight line.





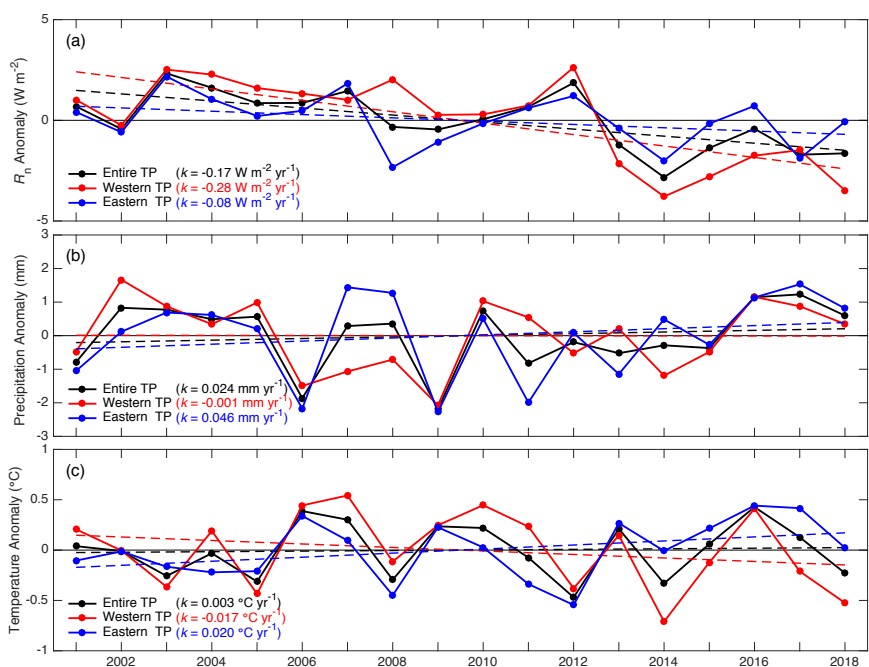


Figure 8: Domain-averaged anomalies of annual $R_n$ (a), precipitation (b), and

temperature (c) over the entire TP, the western TP (lon < 90° E), and the

eastern TP (lon > 90° E), respectively. The dashed straight lines indicate

linear trends during 2001-2018, and $k$ is the slope of the straight line.

639