# Peer review of "Long term variations of actual evapotranspiration over the Tibetan"

_Earth System Science Data, 2020_

## Author Comment (AC1)

Response to Referees' Comments:

We would like to thank the editor, the topical editor, and the anonymous referee for the time and efforts handling and reviewing our manuscript. The constructive comments and suggestions are very helpful to improve our manuscript.

The referee's original comments are formatted in black, while our point-by-point responses are formatted in **blue** font. All the corresponding revisions in the revised manuscript are indicated in **red**.

**Referee 1**

General comments

The authors did an interesting and challenging research, as it is known that the ET is essential to water and energy cycle on the TP. The authors estimated the ET from 2001 to 2018 by using SEBS model, which has contribution to understand the water cycle on the TP, while the MODIS products should be used carefully over the TP for its complicated weather and underlying surface. The authors, please, add more information in detail of the MODIS data used in this study.

We would like to thank the reviewer for the helpful comments and suggestions and for recognizing the contributions made by this work. More detailed information on MODIS data has been added in section 2.2 "Data" in the revised manuscript.

"……*MODIS monthly land surface products, including land surface temperature and emissivity, land surface albedo, and vegetation index, provide land surface conditions for the SEBS model. Detailed information on MODIS land surface variables are listed in Table 1. The values of land surface variables in the MODIS monthly products are derived by compositing and averaging the values from the corresponding month of MODIS daily files. Validations of MODIS land surface temperature and albedo against in-situ observations on the TP suggesting a high quality of MODIS land surface products with low biases and small root-mean-square errors (Wang et al., 2004; Ma et al., 2011; Chen et al., 2014)……*"

And, in this investigation, the estimated ETa by using the SEBS model were validated with six flux tower data from EC observation. It could show the estimated ETa was reasonable used over the whole TP, however, the results are not fully convinced, and the Rn, air temperature and velocity should be evaluated by using the observation data at 6 sites, and then extend to analyze the variations of ET in the western TP ,Eastern TP and the whole TP.

Thank you very much for your comments and suggestions. Validations of the meteorological variables of the China Meteorological Forcing Dataset (CMFD) against in situ observations at the six sites have already been made by several researchers, for

example Wang et al. (2020) and Xie et al. (2017). We also compared meteorological variables including air temperature, specific humidity, wind velocity, downward shortwave, and longwave radiation between CMFD and in situ measurements at the six sites. Please check Figures S1-S6 in the supplementary materials. However, we decided not to discuss the validation of the CMFD dataset in this manuscript, while only focused on the validation of $ET_a$. We added a sentence *"…CMFD dataset has been validated against in situ meteorological observations and compared with other reanalysis datasets on the TP, demonstrating that it is one of the best meteorological forcing datasets over the TP area ...*" in the revised manuscript mentioning the validation of CMFD on the TP to make the current work more convincing.

In general, I would like to recommend accept this manuscript after minor revision and publish it in this journal.

Specific comments

P2: line 37-38," The domain mean of annual ETa on the TP decreased slightly …"should give a reference.

Thank you very much for your comment. This statement is one of the conclusions of this study, and we do not think it is necessary to add a reference.

P3: line 57-58, the sentence does not make sense, please make it clearly.

The sentence has been changed to "…*The SEBS-estimated monthly $ET_a$ during 2001-2018 has been validated against 6 flux towers on the TP…*"

P4:line 78, ET and its variations have been drawing more attention worldwide.

Thank you very much for your suggestion. The sentence has been changed in the revised manuscript.

P5: line 95-102, the authors listed three studies, how does their results performance? Do the authors compare the results (Ma, 2019) with the 6 flux tower observations? It is not clear why the authors conducted this research, just like the authors mentioned, Ma, et al, 2019 they got the Eta from 1982-2012. We know the two methods are both used to estimate ETa at regional and global scale.

Thanks very much for your comments and suggestions. We have not got chance to validate $ET_a$ estimates by Zhang et al., 2018, Ma et al., 2019, or Wang et al., 2020, due to no access to their data. It would be very interesting to compare their results with the 6 flux tower observations.

However, in this study, our intention is not to evaluate the performance of different methods, either based on complementary relationship or surface energy balance. One of the key points of this study is that an improved parameterization scheme for effective aerodynamic roughness length was introduced into the SEBS model, that takes into account not only the shear stress imposed by canopy but also the form drag generated by large-scale topography, which is very important in the mountainous areas of the Tibetan Plateau. Our model is more reasonable physically and $ET_a$ estimate is of high accuracy compared to in situ observations.

P7: line148-149, the "the", in the net radiation flux, the latent heat flux, the sensible heat flux, the ground heat flux, should be removed.

"the" has been removed in the revised manuscript.

P7 and P8, line 169-196, whether the parameter, d0,Cp  and U * are also from CMFA?

Those parameters are not from CMFD. Cp is the specific heat for moist air, and a constant was used. d0 is zero-plane displacement height and u* is fraction velocity, those two variables are parameterized in the SEBS model.

P11, line 272- 273, the specific data should be used to show how does the SEBS performance well at the two sites.

Correlation coefficient and MB value have been used to show the performance of SEBS model is well at the two sites. The sentence has been changed to "……*Specifically, the SEBS model performed exceptionally well at the short grass sites (BJ and NAMORS), with correlation coefficients as high as 0.98 and MB values below 5.0 mm mo-1…*"

P25/P27: The length-width ratio of Figure 1 and other Figures is different.

The length-width ratio of figure 1 has been changed to the same as the rest of figures.

**Reference:**

Wang, B., Y. Ma, Z. Su, Y. Wang, W. Ma. 2020. Quantifying the evaporation amounts of 75 high-elevation large dimictic lakes on the Tibetan Plateau. *Science Advances* 6(26): eaay8558.

Xie, Z., Z. Hu, L. Gu, G. Sun, Y. Du, X. Yan. 2017. Meteorological Forcing Datasets for Blowing Snow Modeling on the Tibetan Plateau: Evaluation and Intercomparison. *Journal of Hydrometeorology* 18(10): 2761-2780.

---

## Author Comment (AC2)

Response to Referees' Comments:

We would like to thank the editor, the topical editor, and the referee Prof. Massimo Menenti for the time and efforts handling and reviewing our manuscript. The constructive comments and suggestions are very helpful to improve our manuscript.

The referee's original comments are formatted in black, while our point-by-point responses are formatted in **blue** font. All the corresponding revisions in the revised manuscript are indicated in **red**.

**Referee 2:**

This study describes a data set on estimates of monthly evapotranspiration on the Tibetan Plateau for 2001-2018. The estimates are based on SEBS and a diverse input data including both satellite retrievals and re-analysis data. The estimates have been evaluated against measurements acquired by six eddy-covariance systems located on the Tibetan Plateau. The dataset is freely available

We would like to thank the reviewer for the helpful comments and suggestions and for recognizing the contributions made by this work.

General comments

The Authors have to be commended for the effort to generate, evaluate against in-situ measurements and make available a potentially useful data set on ET. On the other hand the manuscript has a number of shortcomings, detailed below, which should be addressed prior to publication.

The major issues are:

1) The entire study and the applied model as described seem to assume that the only relevant water phase transition is liquid to vapour, either as evaporation or transpiration. Clearly, on the TP all water phase transitions occur at different times and place and should be taken into account to produce monthly estimates of ET, or better LE. Additional comments are given below.

Thank you very much for your comments and suggestions. We agree with you that water phase transition is taking place all time and everywhere, and is of great importance to the evapotranspiration, water and energy cycle on the TP. We also noticed that this is the main concern of the reviewer Prof. Menenti. The SEBS model we used in this study, is a remote sensing based surface energy balance scheme for estimation of land surface turbulent heat fluxes. The sensible heat flux is calculated based on the atmospheric surface layer Monin-Obukhov Similarity Theory (MOST) and incorporated a variety of parameterization of roughness lengths for momentum and heat transfer, ground heat flux, etc. Latent heat flux or actual evapotranspiration is calculated

as a residual term of the surface energy balance equation considering energy and water limits after estimation of net radiation, ground heat flux, and sensible heat flux. In this study, one of the key points is that an improved parameterization scheme for effective aerodynamic roughness length was introduced into the SEBS model, that takes into account not only the shear stress imposed by canopy but also the form drag generated by large-scale topography, which is very important in the mountainous areas of the Tibetan Plateau. Unfortunately, the water phase transition is not included in current version of the SEBS model. For example, processes related to energy releasing and consuming during freezing and thawing of permafrost and glaciers, and ice sublimation are not considered. We agree that this is a shortcoming of the SEBS model and of this study. Moreover, we have added a sentence to note that the shortcoming in section 2.1 "Model description" in the revised manuscript, which is "……*Note that this equation neglected energy stored in the canopy, energy consumption related to freeze-thaw processes of permafrost and glacier, etc. Thus, this equation is applicable without considering the phase change of water. ……"*

2) Temporal sampling of the data set is not explained and, as a matter of fact, model description does not even mention any temporal dimension. How are monthly estimates actually obtained?

Temporal resolutions of the input data are listed in Table 1. The input data for the SEBS model are at a monthly scale, and as a result, the estimated $ET_a$ is monthly. Moreover, we have introduced how input data are prepared and averaged into monthly variables in the last paragraph in section 2.2 "Data" in the revised manuscript.

"…… *3-hourly CMFD data was averaged into daily and then into monthly data to be consistent with MODIS products in terms of temporal resolution. Daily land surface albedo has been averaged into monthly variable. MODIS land surface products and canopy height data were remapped onto CMFD's grid. Monthly EC data and in situ meteorological observations, which are used for model validation, were generated from half-hourly variables ……"*

3) There are multiple instances of poorly explained land surface processes related to the water phase transition.

As we replied to your first question, due to the limits in the SEBS model, processes related to the water phase transition are missing in this study. It would be very interesting to include freeze-thaw processes of permafrost and glaciers, and sublimation on ice- or snow-covered surfaces, and we would like to look into this issue in the future. Moreover, we have added a sentence to note that the shortcoming in section 2.1 "Model description" in the revised manuscript, which is "……*Note that this equation neglected energy stored in the canopy, energy consumption related to freeze-thaw processes of permafrost and glacier, etc. Thus, this equation is applicable without considering the phase change of water. ……"*

Specific comments

L29 The Abstract should list input data applied.

We have mentioned that meteorological forcing data, satellite products, and in-situ eddy-covariance observations are used in the abstract. We do not think it is a good idea to give detailed information on input data in the abstract.

L71 Not clear whether it is really meant that ET supplies 2/3 of P, in which case the obvious question would be where does the remaining 1/3 come from. This sentence may also mean something completely different, i.e. that land ET is 2/3 of P overland, implying that the remaining 1/3 may come from oceans. Please clarify

Thanks very much for your comment. It is a typo, not meaning ET supplies two-thirds of precipitation, it means that terrestrial ET consumes about two-third of total global terrestrial precipitation. "Contributing" has been replaced by "Consuming" in the revised manuscript.

L75 modulating in which sense?

It means that ET modulates weather and climate via the exchange of water and energy between the atmosphere and ground surface.

L76 "only connecting component" not correct: all water phase changes involve large energy exchanges.

"the only connecting component" has been changed to "one essential connecting component".

L93 "according to contrasting trends 93 between Epan and actual ET…" This is all very confusing, since Epan and ETa cannot be compared. In addition Epan is largely affected by the configuration of the pan. There is an old FAO Technical Bulletin dedicated to the measurement of pan evaporation, which describes in detail how the design of a pan affects the measurement.

We agree that $E_{pan}$ and $ET_a$ are very different. However, studies have reported that $ET_a$ and $E_{pan}$, or $ET_p$ (potential evapotranspiration) exhibit a complementary relationship that means based on variation information of $E_{pan}$ and $ET_p$, one could predict the trend in $ET_a$ (Zhang et al., 2007). Complementary relationship based models have been employed widely to estimate the terrestrial evapotranspiration (Szilagyi et al., 2017). There are also some works that use $E_{pan}$ to estimate $ET_a$ (Sumner and Jacobs, 2005).

L94 this sentence is misleading, since it suggests that Epan may provide information on Eta

As discussed in the response to the last comment, $E_{pan}$ does provide information on $ET_a$ and the sentence remains unchanged.

L107 "plateau-scale variations of ETa" Not quite sure I understand this sentence. To measure spatial variability and patterns a number of measurements at different locations are needed, so what is wrong with a limited footprint of each measurement.?

It means that due to the limited footprint of EC tower, covering the entire TP requires a large amount of EC towers, which is unrealistic.

L121 Model description does not include any information about parameterizations applying to snow and ice.

Processes on ice- and snow-covered surfaces are missing in the SEBS model and this study, which is a shortcoming of this study.

L143 "2.1 Model description" On the TP snow and ice cover a large area, with snow cover varying rather rapidly in time. There is not a single comment about this and SEBS as described does not account for energy and mass exchanges between snow / ice and the atmospheric boundary layer. Moreover, the transition liquid to vapour is not the only one determining the water mass and surface energy balance.

As we replied to your first question, due to the limits in the SEBS model, processes related to the water phase transition are missing in this study. It would be very interesting to include freeze-thaw processes of permafrost and glaciers, and sublimation on ice- or snow-covered surfaces, and we would like to look into this issue in the furture. Moreover, we have added a sentence to note that the shortcoming in section 2.1 "Model description" in the revised manuscript, which is "……*Note that this equation neglected energy stored in the canopy, energy consumption related to freeze-thaw processes of permafrost and glacier, etc. Thus, this equation is applicable without considering the phase change of water. ……*"

L145 "remote-sensed land surface energy" at which time interval? no mention in this entire Section of temporal coverage and sampling.

Temporal coverage and sampling have been given in the last paragraph in section 2.2 "Data" in the revised manuscript.

L149 "latent heat flux" L is different for melt, sublimation and of opposite sign for condensation, freezing and deposition. Is this taken into account?

Processes related to the water phase transition are missing in this study, which is a shortcoming of this study. We have noted this in the last sentence of section 2.1 "Model description" in the revised manuscript.

*"……Note that this equation neglected energy stored in the canopy, energy consumption related to freeze-thaw processes of permafrost and glacier, etc. Thus, this equation is applicable without considering the phase change of water……"*

L164 "Over water surfaces (NDVI < 0 and a < 0.47)" Implications of this sentence not clear

It means over large open water surfaces, for example, lakes.

L166 "G0 is negligible" This is a very peculiar statement, since it is the heat absorbed by a glacier that drives melt. Also, the thermal conditions of glaciers are far from stable, even though temperature may remain < 0.

Thanks very much for your comments. We agree that $G_0$ is also important over glacier surfaces. Unfortunately, freeze and thaw processes in glacier are missing in this study, and very simple assumptions are used over glacier surface.

L187 Eq.7 is h the same as z in the previous equations?

h is the mean roughness obstacle height, which is different from z. We added a sentence "h is the average height of the subgird-scale roughness obstacles" in the revised manuscript.

L190 rather unlikely that a DTM can capture roughness elements like rocks and similar details.

DEM data used in this study is to parameterize the effective roughness length, that is to account for the drag force induced by large-scale topography. Our intent is not to capture the small-scale rocks etc.

L198 "an evaporative fraction" What is this supposed to mean? there will always be an evaporative fraction, regardless of whether a model is applied to estimate it and even less relevant is which model is applied.

We intended to mention the dry and wet limiting conditions used in the SESB model. To avoid misleading, the sentence has been changed to "……To constraint the actual evapotranspiration, the evaporative fraction was applied in the SEBS model, which is determined by taking energy balance considerations at dry and wet limiting cases……". Moreover, we also defined the evaporative fraction ($\Lambda$) as Equation (9) in the revised manuscript.

L199 These sentences denote a poor understanding of the fundamentals of energy balance at the limiting conditions.

We have defined the evaporative fraction ($\Lambda$) as Equation (9) in the revised manuscript. The sentences and words have been changed to make a clearer description on the

energy balance considerations at dry and wet limiting cases. Please check the last paragraph of section 2.1 "Model description" in the revised manuscript.

L202 "evaporative fraction" The evaporative fraction is not the residual of the surface energy budget.

"evaporative fraction" has been deleted and the sentence has been changed to "……Latent heat flux was calculated as a residual of the surface energy budget equation accounting for dry and wet limits……"

L207 "CMFD" add full denomination the first time used

CMFD is the abbreviation of the China Meteorological Forcing Dataset, and it has been introduced in Line 184 in the revised manuscript (tracked version) when it is used for the first time.

L211 which data set?

"The dataset" has been changed to "The CMFD dataset".

L236 "post-processing" of what?

The sentence has been changed to "….The main post-processing procedures of the EC raw data …."

L240 "the energy
L241 "storage in the layer above" In snow and ice the latent heat of fusion should be added

The processing is to adjust in situ ground heat flux measurements, which is at a depth of 10 cm below the ground surface. The energy storage between ground surface and the soil heat flux plate cannot be ignored and should be added to the ground heat flux. We agree that the energy related to fusion of snow and ice should be taken into account in the surface energy balance equation, however, we believe that it should not be added to the ground heat flux.

L264 "phase" I guess this depends more on whether the sampling interval is short enough to capture significant fluctuations in the ET signal. Do not see how this might be inherently related to a model.

"phase" has been changed to "seasonal variation" in the revised manuscript.

L282 "mean" does this mean the average of annual ET between 2001 and 2018?

Yes. To make it clearer, the sentence has been changed to "……There was a clear spatial pattern to the multiyear average of annual $ET_a$ between 2001 and 2018 ……"

L292 "have less available energy to evaporate." But snow and ice will melt....

Yes, we agree that melting snow and ice also contribute to evaporation. However, in general, the net radiation on a surface covered with snow and ice is relatively low due to a high albedo of snow and ice.

L295 "evapotranspirated" even in case the approach described would be applicable to any surface including snow and ice, the estimated LE would relate to the net latent heat balance. i.e. to the net energy absorbed and released by all water phase changes, not just to evaporation and transpiration.

Thanks very much for your comments. In section 2.1 "Model description", we have made it clear that in this study energy consumption related to water phase change is not taken into consideration. Thus, the "evapotranspiration" only includes evaporation and transpiration of liquid water.

L303 "Note that the distribution pattern almost faded out in winter season" faded out in which sense?

It means the spatial contrast in winter is not as strong as other seasons. To make it clearer, "distribution pattern" has been changed to "spatial contrast of $ET_a$"

L349 define amplitude and changing rates, otherwise rather ambiguous in this section on trends.

The changing rate has already been introduced in equation (10) in section 2.3 "Model evaluation metrics and data analysis methods". We implied the least-square regression technique to detect the long-term linear changing trend and the slope of the linear equation is defined as the changing trend.

L356 "decrease of Rn" This deserves more attention, since it might be related to increasing albedo as due e.g. to increasing snow cover.

Thanks very much for your suggestions. The decrease of $R_n$ on the TP is indeed an interesting topic and the reasons for the changes in $R_n$ are complicated. As you pointed out it might be related to increasing albedo due to increasing snow cover, and it might also due to the increase in aerosols and clouds that block solar radiation. It is beyond the scope of this study. Thus, we did not discuss the reason why $R_n$ decreased in this manuscript.

L366 "the melting of permafrost and glaciers on the TP. Hence, the melting water." no attention paid, in Model description or anywhere else, to the associated latent heat of melting!

As we replied to your first question, due to the limits in the SEBS model, processes related to the water phase transition are missing in this study. It would be very interesting to include freeze-thaw processes of permafrost and glaciers, and sublimation on ice- or snow-covered surfaces. Moreover, we have added a sentence to note that the shortcoming in section 2.1 "Model description" in the revised manuscript, which is "……*Note that this equation neglected energy stored in the canopy, energy consumption related to freeze-thaw processes of permafrost and glacier, etc. Thus, this equation is applicable without considering the phase change of water. ……*"

L374 "high-spatial resolution" which data sets? is MODIS high resolution? any consideration about length-scale of spatial variability? At L378 10 km is indicated as spatial resolution of the ET estimates, this is by n no means high spatial resolution, especially given the spatial variability of snow and ice.

The MODIS products and meteorological forcing dataset used in the study are definitely not high-spatial resolution. The sentence has been changed to "……This demonstrates the necessity to evaluate the spatial distribution of changing trends in $ET_a$ and utilize long time series to investigate the trends in $ET_a$ over the TP……"

L389 "evapotranspiration" This is a very misleading term to describe all the water phase changes on the TP.

Thanks very much for your comments. To avoid misleading, in section 2.1 "Model description", we have made it clear that in this study energy consumption related to water phase change is not taken into consideration. Thus, the "evapotranspiration" only includes evaporation and transpiration of liquid water.

L407 "rate of change" is this different from the trend mentioned elsewhere?

It is the same as "trend" mentioned elsewhere and it has been changed to "trend" to avoid misleading in the revised manuscript.

L431 "forest canopy height" where used? Model description does not mention canopy height as a relevant variable and does not explain how such data might have been used.

"forest canopy height" has been introduced in table 1, and it was used in the parameterization of aerodynamic roughness length in the SEBS model.

Reference:

Sumner, D. M., J. M. Jacobs. 2005. Utility of Penman–Monteith, Priestley–Taylor, reference evapotranspiration, and pan evaporation methods to estimate pasture evapotranspiration. *Journal of Hydrology* 308(1): 81-104.

Szilagyi, J., R. Crago, R. Qualls. 2017. A calibration-free formulation of the complementary relationship of evaporation for continental-scale hydrology. *Journal of Geophysical Research: Atmospheres* 122(1): 264-278.

Zhang, Y., C. Liu, Y. Tang, Y. Yang. 2007. Trends in pan evaporation and reference and actual evapotranspiration across the Tibetan Plateau. *Journal of Geophysical Research: Atmospheres* 112(D12).

---

## Author Response (AR2)

Response to Referee's Comments:

We would like to thank the editor, the topical editor, and the referee Prof. Massimo Menenti for the time and efforts handling and reviewing our manuscript. The constructive comments and suggestions are very helpful to improve our manuscript.

The referee's original comments are formatted in black, while our point-by-point responses are formatted in **blue** font. All the corresponding revisions in the revised manuscript are indicated in **red**.

general suggestions

In the description of the method it should be made clear that the approach described assumes that only the liquid - vapour phase transition occur and that the energy balance equation is written accordingly, i.e. it is affected by the same limitations. This limitation impacts the interpretation of observed land surface temperature.

The sentence has been added in the revised manuscript to state that only liquid to vapour phase transition has been considered: "……*This equation is not applicable to any condition where a phase change of water occurs, except the liquid to vapour phase change*……"

In the description of the data, potential users should be informed that the data are likely to be less reliable under conditions where other water phase transitions than liquid - vapour may occur. On the TP this is the case at high elevation, in winter and in general anywhere temperatures are below zero.

A sentence has been added to mention the limitation of the dataset in the "Summary and conclusions" section that is "*Note that the energy consumption related to freeze-thaw processes and sublimation is neglected. Thus, the dataset is likely to be less reliable over the glacier, permafrost, and in winter season*."

specific suggestions

L174: the sentence "Thus this equation is applicable without considering the phase change of water" is ambiguous / unclear and should actually read "This equation is not applicable to any condition where a phase change of water occurs, except the liquid to vapour phase change".

The sentence has been changed in the revised manuscript.

L236: here it should be discussed how far the dry – wet limit concept is applicable to conditions, as in the TP, where all water phase transitions occur.

The sentence "……*Note that the dry-wet limit assumption did not apply to frozen soil, water, snow, and ice surfaces. The latent heat flux was obtained as the residual of the surface energy balance equation (1) after calculating net radiation, sensible heat flux, and ground heat flux when the dry-wet limit assumption is not applicable*……." Has been added to describe the limitation of the dry-wet limit assumption.